# Neuroprotection Is in the Air—Inhaled Gases on Their Way to the Neurons

**DOI:** 10.3390/cells12202480

**Published:** 2023-10-18

**Authors:** Stefanie Scheid, Ulrich Goebel, Felix Ulbrich

**Affiliations:** 1Department of Anesthesiology and Critical Care, Medical Center, Faculty of Medicine, University of Freiburg, 79106 Freiburg, Germany; felix.ulbrich@uniklinik-freiburg.de; 2Department of Anesthesiology and Critical Care Medicine, St. Franziskus-Hospital, 48145 Muenster, Germany; ulrich.goebel@sfh-muenster.de

**Keywords:** neuronal cell death, ischemia/reperfusion injury, volatile anesthetics, noble gases, isoflurane, sevoflurane, desflurane, xenon, carbon monoxide, hydrogen sulfide, neon, helium, argon

## Abstract

Cerebral injury is a leading cause of long-term disability and mortality. Common causes include major cardiovascular events, such as cardiac arrest, ischemic stroke, and subarachnoid hemorrhage, traumatic brain injury, and neurodegenerative as well as neuroinflammatory disorders. Despite improvements in pharmacological and interventional treatment options, due to the brain’s limited regeneration potential, survival is often associated with the impairment of crucial functions that lead to occupational inability and enormous economic burden. For decades, researchers have therefore been investigating adjuvant therapeutic options to alleviate neuronal cell death. Although promising in preclinical studies, a huge variety of drugs thought to provide neuroprotective effects failed in clinical trials. However, utilizing medical gases, noble gases, and gaseous molecules as supportive treatment options may offer new perspectives for patients suffering neuronal damage. This review provides an overview of current research, potentials and mechanisms of these substances as a promising therapeutic alternative for the treatment of cerebral injury.

## 1. Introduction

Neuronal cell death is a common pathological finding in various disorders of the central nervous system (CNS), including vascular disease, neurodegenerative disorders affecting the brain and spinal cord, multiple sclerosis, and traumatic brain injury [1,2]. Due to the limited regeneration potential of the central nervous system, affected patients often show protracted disability. Therefore, the development of novel therapeutic strategies that are able to mitigate and potentially reverse neurological deficit is of vital importance [2].

More specifically, neurological dysfunction due to hypoxic brain injury is a leading cause of long-term disability and mortality after major cardiovascular events, such as cardiac arrest and stroke [3]. Stroke that occurs in response to ischemic heart disease is the second most common cause of death and severe disability worldwide. Stroke survival is associated with the impairment of crucial functions that lead to occupational inability and enormous economic burden [4]. The cessation of cerebral blood flow leads to anoxia and a deficient nutrient supply, which damages neuronal cells due to their lack of intracellular energy stores [5]. The restoration of blood flow, then, has several repercussions, including mitochondrial dysfunction, the release of pro-apoptotic proteins and reactive oxygen species (ROS), and activation of the immune system with subsequent tissue inflammation, which can cause secondary reperfusion injury [5]. The two main etiologies of stroke are ischemic and hemorrhagic, with most strokes (87%) caused by ischemic events (i.e., the embolic or thrombotic occlusion of cerebral arteries). In hemorrhagic stroke, arterial hypertension leads to subarachnoid aneurysms that may consecutively rupture to cause devasting intracerebral hemorrhage. Intracranial vascular malformations further contribute to the predisposition to hemorrhagic stroke in this group [6]. Therapeutic options for terminating cerebral ischemia and restoring vascular flow for both stroke types are limited and often not feasible for technical or logistic reasons. In hemorrhagic stroke, the therapy of choice is an immediate interventional or surgical treatment: aneurysm clipping and the repair of arteriovenous malformation are commonly used to control bleeding and avoid increased intracranial pressure. Depending on the localization, coil embolization of ruptured aneurysms represents a good treatment option. In ischemic strokes, intravenous thrombolysis with a tissue plasminogen activator can re-open occluded vessels within a limited time frame [7]. For large-vessel ischemic stroke, mechanical thrombectomy with specifically designed stent-retrieval systems is another safe and effective possibility for revascularization even within 6 h or longer of the ischemic event [8]. Overall, immediate revascularization has absolute priority to stop ischemia and restore oxygen and glucose supplies to avoid further neuron destruction—the secondary injury. Conservative treatments such as oxygen therapy and blood pressure control are also required. In recent years, the outcomes of stroke treatment have not been optimal, raising the following question: what other options might be available for saving or protecting neurons from functional decline and destruction and for preserving neurological function? For decades, researchers have investigated the potential of a huge variety of drugs thought to provide neuroprotective effects [9]. Although promising in preclinical studies, these drugs have failed to prove neuroprotective effects in clinical trials. These drugs ranged from Nox inhibitors, magnesium sulfate, radical scavengers, and antibiotics like minocycline to hematopoietic growth factors and cytokine antagonists [9,10,11,12,13,14,15].

Since the need for effective treatment options is still huge, it is tempting to speculate about volatile agents and medical gases as a possible link to provide distinguished medical treatment for neurological disorders. Volatile agents provide the option to easily treat neuronal injury in a safe and effective manner and have demonstrated positive effects on cerebral recovery after ischemia in experimental settings and clinical feasibility studies, although to date, no outcome data are available to prove a clinically significant neuroprotective effect [10]. In this review, we will discuss the effects of different volatile agents: those that are already present in routine clinical use like isoflurane and sevoflurane, and furthermore potentially advantageous experimental molecules like carbon monoxide, hydrogen sulfide, xenon, argon, or helium, whose beneficial effects have only been recognized in animal-based models of neuronal injury so far [11]. Nevertheless, existing data suggest that all these substances provide certain—but not all-embracing—potential to treat certain neurological conditions in the future.

## 2. Pathophysiology of Ischemia and Reperfusion Injury

In a stroke, different pathophysiological mechanisms lead to the observed neuronal cell death. Cerebral ischemia promotes anaerobic metabolism in neurons and shifts associated cells into an acidotic state [12]. The lack of oxygen and glucose, in combination with impaired membrane permeability, decreases adenosine triphosphate (ATP) production in mitochondria, thereby inactivating the Na^+^/K^+^ ATPase. A resulting dysfunction in the Na^2+^/Ca^2+^ exchanger, in turn, reduces Ca^2+^ efflux from the cell and limits Ca^2+^ uptake into the endoplasmic reticulum, resulting in an overload of Ca^2+^ in the cell [13]. The deficiency of ATP on the one hand and anoxic depolarization at the presynaptic terminal on the other hand also cause dysfunction of glutamate transporters, causing an accumulation of glutamate, an important excitatory neurotransmitter [14], in the synaptic gap and leading to excitability [15]. Glutamate, in turn, activates N-methyl-D-aspartate (NMDA) receptors, further exacerbating the massive Ca^2+^ influx [16]. Neuronal Ca^2+^ overload activates apoptotic proteins like caspases, finally inducing apoptosis [17]. Neuroinflammation constitutes another key contributor to neuronal injury in cerebrovascular diseases, especially in the acute phase of stroke [18]. Vascular occlusion induces the altered expression of cell surface and adhesion molecules in endothelial cells, which leads to attraction, adhesion and transmigration of leukocytes into the ischemic tissue, leading to an increase in tissue inflammation [19]. Ischemia also results in activation of microglia, the resident immune cells of the brain [20].

Neuronal damage can also be induced by recanalization [21], as reperfusion and re-oxygenation after ischemia cause a massive formation of ROS and activation of immune cells. The infiltrating immune cells release inflammatory mediators, including tumor-necrosis factor (TNF)-α, Interleukin-1β, and Interleukin-6, which activate a cascade of signaling pathways that increase brain levels of reactive nitrogen species, neurotoxic molecules and free radicals. These molecules disrupt the blood–brain barrier, thereby generating a secondary brain injury [22]. The mechanism of activation of the inflammatory pathway is complex and is mediated by Toll-like receptors (TLR), which are stimulated by non-microbial ligands, such as damage-associated molecular patterns (DAMPs). Several of these ligands, including high-mobility group box (HMGB)-1, heme oxygenase-1 (HO-1), and Interleukin-6 are released from the cytoplasm after tissue damage and initiate the innate immune response [23].

## 3. Inhaled Gases Providing Protection against Neuronal Ischemia

### 3.1. Volatile Anesthetics: Isoflurane, Sevoflurane, and Desflurane

Isoflurane, sevoflurane, and desflurane are halogenated ether compounds routinely used for the induction and maintenance of general anesthesia.

Sevoflurane has been described as a neuroprotective agent in both experimental and clinical settings. It seems to affect several mechanisms, including inflammation, oxidative stress, and apoptosis, that are implicated in the pathophysiology of stroke. The anti-inflammatory properties of sevoflurane were originally described in the context of myocardial protection from ischemia/reperfusion injury [24]. However, studies in rats have reported that sevoflurane (2% 15 min before reperfusion) decreased both the local and systemic inflammatory reactions after cerebral ischemia/reperfusion injury, resulting in lower levels of pro-inflammatory cytokines such as TNF-α, Interleukin-1β, and Interleukin-6 [25]. At the molecular level, this effect seems to be at least partly mediated by inhibition of the TLR-4/Nuclear Factor Kappa B (NF-κB) pathway (Figure. 1) [25,26]. Isoflurane, but not sevoflurane promotes part of its detrimental action with the p75-NTR receptor, located solely in neurons. Schallner et al. were able to demonstrate that inhibition of p75NTR decreases isoflurane-induced damage to neurons in vitro and in vivo via an NF-kB related pathway [27]. Restin et al. observed that rats treated with sevoflurane (2.2% for 4 h) following hypoxia and reoxygenation injury maintained both vascular endothelial integrity and the brain barrier function, indicating that sevoflurane protected against tissue inflammation [28]. Conversely, sevoflurane (2.2% for 12 h) reduced systemic inflammation but not neuro-inflammation in a rat sepsis model [29].

Oxidative stress due to the increased production of ROS plays a key role in the pathophysiology of stroke. Sevoflurane treatment (0.15–3.4 mM) of rat glial cells was reported to reduce the release of free radicals in a concentration-dependent manner, indicating an anti-oxidant function [30]. Rats provided sevoflurane post-conditioning (2.5% in 30% O_2_ for 30 min) after hypoxia–ischemia also showed improved neuronal survival and decreased apoptosis and cellular atrophy through regulation of the phosphatidylinositol-3-kinase (PI3K)/proteinkinase B (Akt) pathway, whereas the exposure of neonatal rats to low-dose sevoflurane (1.2% for 6 h) increased hippocampal neurogenesis and synaptic plasticity, resulting in improved spatial learning and memory development later in life [31,32]. Post-conditioning refers to a strategy where a particular gas is applied after an ischemic event and reperfusion already took place, thereby modifying reperfusion-induced adverse events, whereas during pre-conditioning, the gas is applied prior to the onset of ischemia. Chen et al. found that sevoflurane (2.7% for 45 min) exerted its neuroprotective effect in part through Akt activation [33]. Furthermore, sevoflurane (2% for two periods of 10 min) reduced apoptosis by increasing phospho-januskinase (P-JAK) and phospho-signal transducer and activator of transcription (P-STAT) expression after transient global ischemia in rats (Figure 1) [34]. In addition, the sevoflurane treatment of rats (2.5% for 90 min) attenuated reactive astrogliosis and glial scar formation after ischemia/reperfusion brain injury, thereby improving neurological function [35]. However, sevoflurane as well as the other volatile anesthetics may exert neurotoxic or neuroprotective effects, depending on the doses administered. As mentioned above, in experiments conducted by Chen et al., the exposure of neonatal rats to low-dose sevoflurane (1.2%), but not higher-dose sevoflurane (2.4%), positively impacted hippocampal neurogenesis and synaptic plasticity, whereas Edwards et al. observed epileptogenic and neurotoxic effects at emergence after the exposure of neonatal rats (postnatal days 4–17) to sevoflurane in a concentration of 2.1% for 0.5–6 h [32,36]. 

As observed with sevoflurane, isoflurane has also shown neuroprotective effects in various experimental studies. Li et al. reported that pre-conditioning with isoflurane (1.5% for 2,4, or 8 h) activated hypoxia inducible factor (HIF)-1 alpha, which protected rat hippocampal neurons against neuronal injury induced by oxygen-glucose deprivation [37]. HIF-1 alpha is a DNA binding complex that enhances the transcription of genes, enabling cell survival after ischemia [38]. Subsequently, Li et al. observed an activation of extracellular signal-related kinases 1 and 2 (Erk1/2) and increased levels of inducible nitric oxide synthase (iNOS) mRNA levels (Figure 1) [37]. In animal experiments, Wang et al. described neuroprotection using isoflurane pre-conditioning (15 mL/L for 1 h over 5 consecutive days) via the upregulation of TWIK-related K^+^ channel (TREK) 1 and attenuation of apoptosis in a rat model of spinal cord ischemic injury [39]. Similarly, Xiong et al. observed a dose-dependent neuroprotective effect of isoflurane pre-conditioning (0.75%, 1.5% or 2.25% isoflurane in O_2_ for 1 h/d over 5 days) that involved an activation of ATP-regulated K^+^ channels after focal cerebral ischemia in rats [40]. Isoflurane pre-conditioning (1%, 2%, or 3% isoflurane for 1 h 30 min prior to an inflammatory stimulus with lipopolysaccharide (LPS) and interferon (IFN)-ɣ) also reduced the activation of murine microglial cells, thereby dampening inflammation and possibly providing neuroprotection [41].

Post-conditioning with isoflurane can also provide neuroprotection after ischemic cerebral injury. Wang et al. reported that rats subjected to cerebral ischemia–reperfusion injury showed a reduced infarct volume and improved neurobehavioral performance following isoflurane post-conditioning (1.5% or 3% for 1 h at reperfusion). The underlying mechanism appeared to involve regulation of the transforming growth factor (TGF)-β signaling pathway and the downstream c-Jun N-terminal kinase (JNK) signaling pathway (Figure 1) [42]. Zhou et al. also observed an isoflurane-mediated (2% for 1 h immediately after hypoxia) reduction in infarct volume (21.9 ± 3.4% vs. 31.4 ± 2.1% with and without exposure to isoflurane, respectively) in a rat model of neonatal hypoxic ischemic injury. This effect was likely mediated by increased production of Sphingosine 1-phosphate, a sphingolipid metabolite that binds with specific G-protein coupled receptors to regulate cell proliferation, survival, migration, and apoptosis and to activate its downstream target, the PI3K/Akt signaling pathway [43]. 

Ischemia/reperfusion damages mitochondria, resulting in increased mitochondrial membrane permeability, enhanced mediator release, and, ultimately, cell damage and death [44]. Li et al. demonstrated that post-conditioning with isoflurane (2% for 1 h at the onset of reperfusion) after middle cerebral artery occlusion (MCAO) in rats attenuated an increase in mitochondrial-membrane permeability, possibly by activation of Akt, a pro-survival protein kinase that modulates mitochondrial-membrane permeability [45].

Although less commonly investigated, there is some evidence that desflurane can provide neuroprotective effects as well. The long-duration exposure to desflurane (8–9% for 24 h) reduced cerebral infarct size by 55% in a rat model of focal cerebral ischemia [46]. This effect was corroborated by Tsai et al., who observed a significantly decreased infarct size after cerebral ischemia/reperfusion injury and sedation with desflurane (minimal alveolar concentration (MAC) 1, 1.25, or 1.5), compared to intraperitoneal chloralhydrate (infarct volume, 206.4 ± 28.5 mm^3^ in animals sedated with chloral hydrate; vs. 132.0 ± 17.6 mm^3^ in animals sedated with 1.0 MAC desflurane, 73.5 ± 23.6 mm^3^ in animals sedated with 1.25 MAC desflurane, and 117.1 ± 29.8 mm^3^ in animals sedated with 1.5 MAC desflurane) [47]. Similarly, desflurane post-conditioning (4.8% at 0.5, 1, or 2 h after hypoxia–ischemia) reduced hypoxic-ischemic brain injury and improved motor function, learning and memory in neonatal rats, possibly by inhibiting transient receptor potential ankyrin (TRPA) 1, an ion channel that functions as a chemical and mechanical stress sensor [48,49]. In accordance with these results, McAuliffe et al. observed a limited neuroprotective effect of delayed pre-conditioning with desflurane (8.4%), isoflurane (1.8%), or sevoflurane (3.1%, for 3 h 24 h prior to hypoxia–ischemia, respectively) against neonatal hypoxia–ischemia in mice [50]. In a murine model of aneurysmal subarachnoid hemorrhage (SAH)-associated delayed cerebral ischemia, post-conditioning with desflurane (6%) or sevoflurane (2%, for 1 h 60 min after SAH, respectively) attenuated large artery vasospasms, reduced microvascular thrombosis, and resulted in improved neurological function [51].

Taken together, pre-clinical data suggest a neuroprotective effect of volatile anesthetics in animal models of stroke (Table 1). Regarding clinical outcomes, in a retrospective analysis of stroke patients undergoing endovascular thrombectomy, general anesthesia with propofol (mean predicted plasma concentration, 2.7 µg/mL) versus volatile anesthetics (desflurane, mean MAC 0.57; sevoflurane, mean MAC 0.61) was associated with improved functional independence after 3 months in the propofol group [52]. On the other hand, a randomized trial in 128 patients undergoing cardiac surgery showed better short-term postoperative cognitive performance after general anesthesia with sevoflurane (mean MAC, 0.6–1.0) compared to propofol (3–5 mg/kg/h) [53]. Notably, due to a lack of authorization to apply sevoflurane during cardiopulmonary bypass (CPB), patients in the sevoflurane group received propofol (3–5 mg/kg/h) for the maintenance of anesthesia during CPB, until the aortic cross-clamp was removed. In another small single-center randomized study performed by Kuzkov et al. in patients undergoing carotid endarterectomy, volatile induction and the maintenance of anesthesia with sevoflurane resulted in improved cerebral oxygenation and early postoperative cognition compared to total intravenous anesthesia (TIVA) with propofol [54]. Conversely, a larger multi-center trial found no difference in 1-year-mortality or adverse cerebral outcomes, including stroke, delirium and postoperative cognitive impairment, in patients receiving general anesthesia with volatile anesthetics (sevoflurane, 83.2%, desflurane, 9.2%, isoflurane, 5.8%, mean MAC, 1.12 before CPB, 0.91 during CPB, 1.02 after CPB) or TIVA (propofol, 87.7%, midazolam, 32.2%; no specific administration protocol implemented; dosages not reported) for elective coronary artery bypass grafting [55]. In contrast, in patients undergoing non-cardiac surgery under general anesthesia, a recent large multi-center retrospective cohort analysis showed that volatile anesthetics (desflurane, isoflurane, or sevoflurane; mean overall MAC 0.73) reduced the incidence and severity of postoperative stroke in a dose-dependent manner, with the strongest effect observed in the early postoperative period and with high-dose volatile anesthetics (highest tertile) [56]. 

In a small retrospective study, patients who received desflurane (1.0–1.5 MAC) for the maintenance of anesthesia during emergent aneurysm clipping had a lower incidence of transcranial Doppler-evident vasospasm compared to patients receiving propofol (target-controlled infusion with a target concentration of 3–4 µg/mL) for the maintenance of anesthesia [57]. In another retrospective analysis on patients with aneurysmal SAH, anesthesia with inhalational anesthetics (sevoflurane or desflurane) only during aneurysm treatment decreased the risk of angiographic vasospasm compared to combined inhalational and intravenous anesthesia, although the end-tidal concentrations of sevoflurane and desflurane did not significantly differ between patients with and without angiographic vasospasm (sevoflurane: 1.28 ± 0.39% in the patient group with angiographic vasopasm vs. 1.34 ± 0.36% in the patient group without angiographic vasospasm; desflurane: 4.4 ± 1.3% in the patient group with angiographic vasopasm vs. 4.9 ± 1.1% in the patient group without angiographic vasospasm), and greater desflurane exposure was associated with less delayed cerebral ischemia (desflurane: 4.0 ± 1.0% in the patient group with delayed cerebral ischemia versus 5.0 ± 1.0% in the patient group with no delayed cerebral ischemia [58].

On another note, while all volatile anesthetics are greenhouse gases and, thus, contribute to climate change, their global warming potential and atmospheric lifetime vary and are much higher for desflurane compared to sevoflurane and isoflurane [59]. Considering the particularly high pollutant effect of desflurane and due to the lack of clinical benefits compared to environmentally friendlier alternatives such as sevoflurane, an increasing number of clinical anesthesia providers avoids its use altogether [60].

In summary, volatile anesthetics may exert neurotoxic or neuroprotective effects, depending on the doses administered. While effects on both sides are usually not seen in clinical routine using low doses, higher doses may trigger beneficial or detrimental effects. Therefore, volatile anesthetics might be beneficial for the prevention and treatment of cerebral ischemia–reperfusion injury during stroke, but available data are inconsistent. Clinical findings observed so far need to be confirmed or ruled out in large randomized controlled trials. Additional research is warranted to determine the optimal dose and timing of the application and to further elucidate the underlying molecular mechanism by which volatile anesthetics exert their neuroprotective effects. From a sustainability standpoint, it can only be recommended that the use of desflurane should be abandoned and additional research should focus on sevoflurane and isoflurane.

### 3.2. Gaseous Molecules

#### 3.2.1. Hydrogen Sulfide (H_2_S)

Although traditionally regarded as a poisonous gas, H_2_S has become recognized as an important signaling molecule with diverse biological functions across several organ systems, including the cardiovascular system and the CNS [61]. Similar to nitric oxide (NO) and carbon monoxide (CO), it functions as a gaseous transmitter [62]. In the CNS, H_2_S is synthesized predominantly by the enzyme cystathionine β-synthase in astrocytes [63].

Organ protective effects of H_2_S have been described in various disease models. More specifically, H_2_S seems to play an important role in neuroprotection. In an in vitro analysis, H_2_S (100 µM sodium hydrosulfide (NaHS)) protected neurons against oxidative stress [64]. Similar results were obtained in an in vivo study, where H_2_S reduced oxidative stress in the mitochondria of fetal rat brains after ischemia/reperfusion injury [65]. The exposure to a mixture of air and H_2_S (80 ppm) for 2 days after ischemia induced profound hypothermia and resulted in a decrease in infarct size by 50% and improved performance on sensorimotor function tests after MCAO, an animal model of focal ischemia, in rats [66]. Concurrently, Wang et al. observed an increase in the blood–brain barrier integrity after cerebral ischemia and treatment with a low dose of the H_2_S donor NaHS (25 µmol/kg/d) or with the slow-release organic H_2_S donor ADT (50 mg/kg), while treatment with ADT also resulted in significantly reduced infarct size and improved neurological function [67]. These results were corroborated by Liu et al., who were able to show that the administration of the H_2_S donors ADT (50 mg/kg)- or NaHS (25 µmol/kg)-attenuated tissue plasminogen activator induced cerebral hemorrhage following experimental stroke in mice [68]. Conversely, higher concentrations of H_2_S can result in neurotoxicity. Qu et al. observed an increase in infarct volume by approximately 50% after MCAO in rats with the administration of 180 µmol/kg, but not with 90 µmol/kg, of NaHS [69]. Mechanistically, H_2_S acts as a regulator of numerous cellular mechanisms that play a key role in the pathophysiology of stroke, such as inflammation, oxidative stress, autophagy and the regulation of cell death [70]. Concordantly, in analyses conducted by Biermann et al. and Scheid et al., pre-conditioning as well as post-conditioning with H_2_S (80 ppm for 1 h before or at 0 and 1.5 h after ischemia/reperfusion, respectively) decreased the levels of pro-inflammatory cytokines TNF-α and Interleukin-1β, reduced the expression of the pro-apoptotic Bax protein and increased the expression of the anti-apoptotic protein BCL-2 in a rat model of retinal ischemia/reperfusion injury (Figure 2) [71]. The neuroprotective effect of H_2_S was at least partly mediated by regulation of the p38/Erk1/2 signaling pathway: Scheid et al. demonstrated that the inhalative as well as intravenous application of H_2_S increased Erk1/2 phosphorylation, and that its protective effect was partly abolished by the Erk1/2 inhibitor PD98059 (Figure 2) [72,73]. In this study, H_2_S treatment also reduced the expression of numerous apoptotic and inflammatory markers, such as caspase-3, intracellular adhesion molecule (ICAM)-1, vascular endothelial growth factor (VEGF), and iNOS [73]. Furthermore, H_2_S is able to alleviate brain edema and cerebral vasospasm after ischemia/reperfusion injury [74].

To summarize, there is growing evidence that H_2_S provides neuroprotection during cerebral ischemia (Table 1). Despite these promising experimental data, H_2_S has not yet been examined in clinical studies. Further research on its significance as a prospective therapeutic seems warranted. Due to concerns regarding storage and handling of the potentially toxic gaseous H_2_S and its unpleasant odor, it might be beneficial to focus on intravenous H_2_S donors for future clinical research. However, its relatively narrow therapeutic window might limit its routine clinical use.

#### 3.2.2. Carbon Monoxide (CO)

Another gaseous molecule has attracted the attention of many researchers in recent decades: CO may offer protection through anti-inflammatory, anti-apoptotic, and anti-proliferative effects [75,76]. In the human brain, endogenous CO is a product of heme catabolism by heme oxygenases (HO), resulting in the generation of Fe^2+^, CO, and biliverdin. The interaction between HO and CO (CO-HO-1 axis) plays a role in a variety of signal transduction mechanisms in the regulation of cell function and communication [77]. Since HO is considered an effective antioxidant in the CNS, the synthesis of the HO-1 subtype is activated during cerebral ischemia–reperfusion injury and is highly concentrated in the border of the infarcted tissue and glial cells [78]. An increase in HO-1 following normobaric hypoxia was also detected in astrocytes, and the consecutive increase in intracellular cGMP in neurons led to a reduced activity of caspase-3 and therefore protected against apoptosis and cell death [79].

Now, the exciting question arises of whether exogenously supplied CO has an additional supportive effect on neuronal tissue. Zynalov and Doré found that inhaled CO in concentrations of 125 to 250 ppm can protect the brain from transient focal ischemia and reperfusion injury: in a mouse model of transient middle cerebral artery occlusion (tMCAO), low levels of CO reduced infarct volume by 32.1 ± 8.9% and 62.2 ± 14.4%, respectively, and cerebral edema [80]. Similarly, in a murine model of SAH, the inhalative application of CO (250 ppm) not only significantly decreased hematoma volume, as well as the incidence of vasospasm and neuronal apoptosis, but was also associated with improved cognitive function after SAH [81]. However, it is well known that inhaled CO in high doses has a toxic effect as a result of binding to oxygen sites [82].

As both CO and H_2_S are potentially poisonous gases, a crucial point in providing CO or H_2_S therapy to a broad range of patients is to avoid patient as well as handler toxicity. Since inhalative delivery of CO requires specific applicators and environmental poisoning must be avoided, expectations of further research are focused on CO-releasing molecules (CORMs) that enable a controlled delivery of micromolar concentrations of CO. Here, the chemical structure of the CORM is decisive for the capability of carrying and liberating controlled quantities of CO without altering COHb to toxic levels [83,84]. The fact that CORMs can be used successfully has been confirmed by Wang et al. CORM-3-treated mice (4 mg/kg body weight) that underwent tMCAO had a significantly smaller infarct volume, lower brain water content and enhanced neurologic outcome. In this study, CORM-3 reduced neuroinflammation as demonstrated via the downregulated expression of ionized calcium-binding adapter molecule (Iba-1), TNF-α and Interleukin-1ß [85]. Concordantly, Schallner et al. observed that the CORM ALF-186 protected neuronal cells from apoptosis and ischemia/reperfusion injury both in vitro and in a rodent model of neuronal ischemia/reperfusion injury [86]. Ulbrich et al. were able to corroborate these results: treatment with ALF-186 (10 mg/kg) abated histological damage, increased the expression of soluble guanylyl cyclase (sGC) ß1, reduced the expression and phosphorylation of NF-κB, modulated the heat-shock response and alleviated neuroinflammation, affirmed by the diminished local and systemic expression of pro-inflammatory markers TNF-α and Interleukin-6, after retinal ischemia/reperfusion injury in rats (Figure 2) [87]. Mechanistically, this effect was mediated by the differential activation of MAP kinases: ALF 186 activated the phosphorylation of p38 and suppressed Erk1/2 phosphorylation [88]. A study by Wollborn et al. found that CO application reaching 7–13% Co-Hb additional to extracorporeal life support (ECLS) after cardiac arrest significantly reduced cerebral damage parameters in histology, molecular biology, and improved functional parameters in neuromonitoring in pigs [89]. Interestingly, in this study, a controlled extracorporeal CO delivery approach was used, avoiding both the risk of overdosage and the potential for toxicity from transition metals associated with inhalative CO application or the systemic application of CORMs [90]. To date, different companies are working on devices to safely applicate CO or H_2_S in different environments.

Apart from the fact that both H_2_S and CO are produced endogenously in the direct reaction to cellular stress, both have relevant concentration-specific (side-)effects, ranging from nausea and vomiting over headache and dizziness to seizure and death. The concentration of CO used in animal or human research are well below any toxic threshold [91,92] and absolutely comparable to heavy smoking—resulting in CO-Hb levels around 8% [83] In contrast, H_2_S has a lower absolute concentration threshold to induce the named side effects. While 80 ppm seems effective at providing neuroprotective effects, concentrations above may cause first unwanted effects while not increasing any protection.

Taken together, although existing preclinical studies regarding CO and ischemia and reperfusion injury are interesting and encouraging (Table 1), based on its reputation as a poison and its relatively narrow therapeutic window, with potentially fatal effects when overdosed, it is highly doubtful whether CO will ever find its way to patients.

**Table 1 cells-12-02480-t001:** Promising volatile gases, gaseous molecules and noble gases providing protection in neuronal ischemia. I/R = ischemia/reperfusion injury, OGD = oxygen glucose deprivation, PrC = preconditioning, PoC = postconditioning.

Substance	Setting	Timing/Dosage	Mechanism	Publication
Sevoflurane	Neuronal glial cells (rat), OGD	3.4 mM Sevo during ODG	Anti-excitotoxic properties	[30]
			via GLT1, decrease in ROS	
	MCAO (rat)	PrC, 2.7% Sevo/97% O_2_	Activation of Akt, GSK3-ß	[33]
		(45 min)	phosphorylation	
	Hypoxia–ischemia brain	PoC, 2.5% Sevo/30% O_2_	PI3K/Akt/eNOS, PI3K/Akt/	[31]
	damage (rat)	(30 min)	GSK3-ß	
	Cerebral I/R (rat)	PoC, 2% Sevo/40% O_2_	JAK-STAT pathway,	[34]
		(2 × 10 min)	reduction in apoptosis	
	MCAO (rat)	PoC, 2.5% Sevo (90 min)	Astrocyte protection,	[35]
			decrease in GFAP,	
			neuro-, phosphoscan	
	Hypoxia/reoxygenation	PoC, 2.5% Sevo/21% O_2_	Endothelial barrier function	[28]
	in endothelial cells (rat)	(4 h)	(VEGF)	
	Cerebral I/R in rats	PoC, 2% Sevo (15 min)	TLR4/NF-ĸB	[25]
Isoflurane	Focal cerebral ischemia (rat)	PrC, 2% Iso/98% O_2_	K_ATP_ channel	[40]
		(1 h/d for 5 d)		
	Hippocampal neurons (rat), OGD	PoC, 2% Iso/95% air (1 h)	Erk1/2, HIF-1α, iNOS	[37]
	Neonatal hypoxia–ischemia (rat)	PoC, 2% Iso/30% O_2_ (1 h)	Sphingosine-1-phosphate/	[43]
			PI3K/Akt pathway	
	MCAO (rat)	PoC, 2% Iso (1 h)	Akt/GSK3-ß pathway	[45]
	Spinal cord ischemia (rat)	PrC, Iso 15 mL/L (1 h/d for 5 d)	TREK1	[39]
	Cerebral I/R (rat)	PoC, 1.5% Iso/85% O_2_ (1 h)	JNK signaling pathway	[42]
Desflurane	Perinatal hypoxic/ischemic brain injury (rat)	PoC, 4.8% Des (1 h)	TRPA1	[48]
H_2_S	Retinal I/R injury (rat) Cellular neuronal injury (rat) Retinal I/R injury (rat) Retinal I/R injury (rat)	PrC, 80 ppm H_2_S (1 h) PrC, NaHS (30 min) PoC, 80 ppm H_2_S (1 h) PoC, 80 ppm H_2_S (1 h)	NF-ĸB, Erk1/2, JNK, HSP-90 PI3K/Akt/Nrf2, ROS reduction NF-ĸB, Akt, Bcl-2, Bax p38/Erk1/2 pathway	[71] [70] [72] [73]
CO	Hypoxia in neuronal cells (human)	PrC, CORM ALF186 (50 µmol/L)	sGC activation	[86]
	Retinal I/R injury (rat)	PoC, CORM ALF186 (10 mg/kg)	p38, Erk1/2, Bax, Bcl-2	[88]
	Retinal I/R injury (rat)	PoC, CORM ALF186 (10 mg/kg)	sGC, NF-ĸB, CREB	[87]
	tMCAO (mouse)	PoC, CORM-3 (4 mg/kg)	Iba-1, TNF-α, IL-1ß	[85]
	Cardiac arrest in pigs	CORM	Iba-1, Caspase-3, HO-1	[89]
Helium	Neonatal hypoxia–ischemia model (rat) Neonatal hypoxia–ischemia model (rat) Neonatal hypoxia–ischemia model (rat)	PrC, 70% He/30% O_2_ (3 × 5 min) PrC, 70% He/30% O_2_ (3 × 5 min) PrC, 70% He/30% O_2_ (3 × 5 min)	Inhibition of apoptosis iNOS, Nrf-2, SOD-1, HO-1 Inhibition of inflammation	[93] [94] [95]
Xenon	Patch-clamp recordings, cell culture Primary neuronal cell culture, OGD/ Hypoxic–ischemic injury (mouse) Patch-clamp recordings, cell culture tMCAO (mouse)	- PrC, 70% Xe/30% O_2_ (2 h) - PrC, 70% Xe/30% O_2_ (2 h)	Activation of TREK-1 BAX, BCL-2 Inhibition of NMDA receptor Increase in K_ATP_ currents HIF-1α	[96] [97] [98] [99] [100]
Argon	Retinal I/R injury (rat)	PoC, up to 75% Ar/21% O_2_ (1 h)	NF-ĸB, Caspase-3	[101]
	Cellular neuronal injury (human)	PoC, 75% Ar/21% O_2_ (1 h)	Erk1/2	[102]
	Retinal I/R injury (rat)	PoC, 74% Ar/21% O_2_ (1 h)	TLR-2, TLR-4	[103]
	Neuronal cell culture, neonatal hypoxia–ischemia brain injury (rat)	Cells: 75% Ar/20% O_2_ (2 h) Rats: 70% Ar/30% O_2_ (2 h)	HO-1, PI3K/Akt, NF-ĸB	[104]
	Retinal I/R injury (rat) Retinal I/R injury (rat)	PoC, 75% Ar/21% O_2_ (1 h) PoC, 75% Ar/21% O_2_ (1 h)	TLR-2, TLR-4, Interleukin-8 Cytokine reduction	[105] [106]

### 3.3. Noble Gases

#### 3.3.1. Neon

The noble gases neon, helium, xenon and argon have all been attributed protective effects over a range of organs and disease models. While some of them have been extensively studied, data on neon are relatively scarce. Neon is the second-lightest noble gas and shows no anesthetic effect under normobaric conditions.

Pre-conditioning with neon (70% for 3 cycles of 5 min before LAD occlusion) reduced myocardial infarct size in rabbits, exerting a similar cardioprotective effect compared to helium, argon, and ischemic preconditioning [107]. In regard to neuroprotection, neon (50% for 24, 48 and 72 h after injury), contrary to xenon and argon, was devoid of a neuroprotective effect in an in vitro model of traumatic brain injury (TBI) [108]. Similarly, in another in vitro model of neuronal injury provoked by oxygen and glucose deprivation, neon (75% for 24 h after injury) did not provide neuroprotection [109]. Concordantly, neon and krypton showed no effect on hypoxic–ischemic brain injury in a rat hippocampus in vitro, whereas xenon and argon were able to prevent injury (concentration of noble gases, 0.5 atm in a hyperbaric chamber for 24 h after injury, respectively) [110]. 

Taken together, based on the currently available data, the potential of neon as a future adjuvant treatment option for stroke is questionable, while the molecular mechanism remains widely unclear.

#### 3.3.2. Helium

Similar to neon and argon, the noble gas helium lacks an anesthetic effect at ambient air pressure, making it a suitable substance for the treatment of stroke, where the ability to assess neurological function is crucial. Helium has been investigated in the context of various disease models. Due to its physical properties, including its high viscosity and lower density than oxygen, it has been used to reduce airway resistance and respiratory distress in patients with chronic obstructive pulmonary disease, although to date no conclusive evidence has confirmed a clinical outcome benefit [111]. 

Despite its physical inertness, helium has shown clear neuroprotective and cardioprotective effects following pre- and post-conditioning treatment [112]. The potential neuroprotective effect of helium seems to depend on its dosage and time of application. Late pre-conditioning with 70% helium significantly decreased the infarct ratio (infarct ratio, 15.8 ± 4.9% in the hypoxia/ischemia group vs. 3.6 ± 1.1% in the helium preconditioning group), increased the number of viable neurons, and improved sensorimotor and cognitive function in a rodent model of neonatal hypoxia/ischemia-induced injury [93]. Conversely, a combination of pre- and post-conditioning with 70% helium in a rat model of cardiac arrest seemed to decrease apoptosis in the brain but had no effect on neurocognitive function [113]. Late post-conditioning with 70% helium in 30% O_2_ at 2 h after MCAO in rats significantly reduced brain infarct volume (infarct volume, 36.0 ± 17.0% of the involved hemisphere in the control group, vs. 4.0 ± 2.0% in the heliox group) and improved neurological function [114]. By contrast, post-conditioning with 50% helium for 24 h had no clinical or histological neuroprotective effect in a rodent model of cardiac arrest [115]. At the molecular level, NO seems to play an important role in facilitating the neuroprotective effects of helium (Figure 3). Li et al. examined the effect of pre-conditioning with 70% helium in a rat model of neonatal hypoxia–ischemia and found significantly increased NO levels, as well as the elevated expression of some anti-oxidases (e.g., nuclear factor erythroid 2-related factor 2 [Nrf-2], HO-1 and superoxide dismutase-1 [SOD-1]), a reduced brain infarct size, and improved neurological function after helium pre-conditioning. This effect was markedly reduced by inhibiting NO synthase [94]. Furthermore, in the same experimental model, pre-conditioning with 70% helium for 3 intervals of 5 min reduced the brain expression of the inflammatory markers TNF-α und Interleukin-1β, increased the brain expression of neurotrophic and growth factors, promoted brain angiogenesis, and led to improved neurobehavioral outcomes [95]. 

In summary, the available data suggests that the potential neuroprotective effect of helium seems to depend crucially on the dosage and timing of the application (Table 1). 

The clinical use of helium in patients with chronic obstructive pulmonary disease means that ventilator systems that allow the application of helium are now commercially available. Its lack of an anesthetic effect and excellent hemodynamic stability seem to make helium a promising option in the treatment of stroke. However, in experimental studies helium has only been effective for neuroprotection when delivered at high doses (70%). Therefore, patients suffering from end-stage respiratory disease who constantly need oxygen support in high concentrations may not be eligible to receive high concentrations of helium. Furthermore, large animal studies investigating the neuroprotective effect of helium are currently lacking, making its translation to clinical use for neuroprotection still only a remote possibility.

#### 3.3.3. Xenon

The known pathophysiology of stroke makes the application of the noble gas xenon as a neuroprotective agent in stroke an entirely reasonable approach. Xenon, as a noble gas with a full outer electron shell, is chemically inert; therefore, it can interact with surrounding molecules and mediate neuroprotection. The effects of xenon are well studied and xenon is dose-dependently able to exert certain neuroprotective functions. Apart from this, xenon is able to block NMDA receptors, a subtype of glutamate receptor, by competing with the glycine binding site (Figure 3) [98]; whether these effects are related causally remains unknown [116].

Other molecular targets of pharmacological interest have also been identified as having potential neuroprotective effects. For example, xenon can activate K^+^ channels (TREK-1), thereby causing hyperpolarization of the membrane potential and a reduction in the activation threshold [96]. Xenon also functions as a K^+^ channel opener by acting directly on the Kir6.2 subunit, which reduces the ATP inhibition and increases K_ATP_ currents [99].

Xenon treatment also reduces the expression of pro-apoptotic genes, such as BAX, and activates anti-apoptotic proteins, such as BCL-2, ultimately leading to the suppression of apoptosis [97]. Xenon also shows anti-inflammatory effects and upregulates HIF-1 alpha, which may exert cytoprotective effects and reduce neuronal dysfunction (Figure 3) [100]. 

Xenon is used as an anesthetic gas in humans. Its application is considered safe and free of toxic side effects on other organ systems and is characterized by good hemodynamic properties [117,118]. Since the key factors in neuronal injury are the overactivation of NMDA receptors and subsequent Ca^2+^overload in neurons, the neuroprotective effect of xenon seems reasonable and has been successful in preclinical studies. Chakkarapani et al. demonstrated that an inhaled concentration of 50% xenon induced histological neuroprotection in a neonatal asphyxia model in newborn pigs; the effect was especially evident in combination with hypothermia [119]. Treatment with xenon (70%), in combination with hypothermia, also improved the neurological outcome of pigs subjected to circulatory arrest and prolonged resuscitation by reducing hippocampal astrogliosis and the inflammatory response of the putamen [120]. The combined treatment of xenon (70% for 1 h) and mild therapeutic hypothermia (33 °C for 16 h) also preserved neurological function after prolonged cardiac arrest in pigs [120]. These promising results have encouraged researchers to pursue the clinical application of xenon. 

Arola et al. confirmed the safe use of xenon (target concentration of at least 40%, in combination with hypothermia (33 °C) for 24 h) after out-of-hospital resuscitation of adults in a small clinical trial and found at least no negative effects compared to the control group, but a much more positive effect on the cardiovascular system. However, the authors did not discuss neuronal outcome in detail [121].

Two other studies, with neonates after asphyxia and adults after out-of-hospital resuscitation, confirmed the safe use of xenon (30% for 72 h and 40% for 24 h, respectively) in combination with therapeutic hypothermia, but failed to demonstrate any effective neuroprotection for different reasons [122,123]. Multiple preclinical models have investigated the neuroprotective effects of xenon, and the mechanisms are well described (Table 1); clinical trials are ongoing, and its clinical use as a treatment for patients with stroke seems plausible. However, despite its apparent benefits, xenon has some disadvantages. 

First, xenon is expensive due to production costs, and its application as a special vapor prevents its routine clinical use [124]. In fact, its application is only feasible using closed-circuit anesthesia machines with special application units. Second, xenon is an approved anesthetic, making its clinical use unfavorable in patients with stroke, where maintenance and variation of the neurological function under a specific therapy are indispensable for neurological evaluation. Third, xenon antagonizes 5-hydoxytryptamine type (5-HT)-3 receptors associated with nausea and vomiting and therefore may aggravate this side effect (PONV) [125]. In addition, it has to be taken into consideration that the concentrations of xenon used to achieve neuroprotective effects range up to 75%, raising the question of clinical practicability when patients require higher fractions of inspired oxygen. However, studies performed by Campos-Pires et al. using a mouse model of TBI showed that xenon dosages from 30–75% were effective in the treatment of TBI, suggesting some flexibility for necessary increases in oxygen concentration for patients with respiratory disease [112,126].

#### 3.3.4. Argon

Given the known disadvantages of xenon, neuroscientists continue to search for a magic spray for use as a stroke treatment. Recently, argon has attracted attention, as it is also a noble gas, but it has no anesthetic effect under normobaric conditions. Therefore, neurological evaluation is feasible in patients treated with argon. 

Preclinical studies in different rodent models have confirmed argon’s neuroprotective effects. For example, Ulbrich et al. demonstrated that argon postconditioning (argon 75 Vol%, O_2_ 21 Vol%, 2 h) after retinal ischemia–reperfusion injury showed the dose- and time-dependent protection of neuronal ganglion cells by suppressing pro-apoptotic protein expression [101]. Zhao et al. found that argon (70 Vol%, O_2_ 30 Vol%, 2 h), in combination with hypothermia, diminished neuronal cell death, inflammation, and brain infarction volume in neonatal rats after unilateral common carotid artery ligation and subsequent cerebral hypoxia [104]. Ma et al. assessed the extent of argon-mediated neuroprotection by prolonging the observation window in a model with focal cerebral ischemia. After the induction of stroke using permanent vascular occlusion (MCAO), they administered argon 70% by inhalation for 24 h. Compared to the control group receiving nitrogen, the argon-treated group showed a significantly improved neurological outcome, overall outcome, and decreased infarct volume (argon vs. nitrogen; total infarct volume: 161.0 ± 75.0 mm^3^ vs. 302.0 ± 61.0 mm^3^; cortical infarct volume: 104.0 ± 60.5 mm^3^ vs. 194.0 ± 41.0 mm^3^; subcortical infarct volume: 58.0 ± 21.0 mm^3^ vs. 109.0 ± 29.0 mm^3^). Animals that received argon after a delay of 2 h after the start of induction and animals that underwent postischemic reperfusion also showed improved neurological outcomes and recovery. However, the latter two groups showed no difference in mortality or infarct size when compared with the nitrogen controls [127]. Fumagalli induced cardiac arrest in pigs and administered argon (50% or 70 Vol%) during the 4 h postresuscitation period. The small group size prevented the determination of any difference in overall survival, but the neurological recovery was significantly better in the argon group, especially at the higher argon concentration. Argon treatment also significantly reduced neuronal degeneration in the cortex and the activation of reactive microglia in the hippocampus. In addition, the animals showed better blood pressure parameters and faster myocardial recovery after argon treatment. The ventilation parameters after argon inhalation were the same in both groups [128]. The neuroprotective effects of argon were unknown for a long time but are now being decrypted. Argon, as a noble gas, has a full outer electron shell, so it reacts weakly with other molecules, mainly by dipole–dipole interactions and van-der-Walls forces. Nevertheless, argon shows potent and verifiable neuronal effects. Goebel et al. observed that argon (75 Vol% for 2 h after injury) attenuates microglial activation and the release of pro-inflammatory cytokines after retinal ischemia/reperfusion injury in rats [106]. Ulbrich et al. demonstrated time- and dose-dependent anti-apoptotic effects of argon elicited by inhibiting the expression of NF-ĸB, a key regulator of apoptosis and inflammation, and caspase-3 cleavage (25, 50, or 75 Vol% argon for 1 h at 0, 1.5, and 3 h after injury) [101,129]. Consequently, argon diminished ischemia/reperfusion injury, whereas a differentiated induction of the heat shock response became obvious [104]. The initial attempts to elucidate the mechanistic features, as determined for xenon, failed [130]. However, Ulbrich et al. have recently found evidence that TLR-2 and -4 are responsible for argon-mediated neuroprotective effects in vivo and in vitro, as the experimental blocking of TLR-2 and/or TLR-4 partly or fully abolished the beneficial effects of argon (Figure 3) [103]. Notably, the expression of Interleukin-8 was directly affected by the inhibition of the TLRs [105].

Zhao et al. showed that argon combined with hypothermia increased the expression of HO-1 and BCL-2 in the cortex and hippocampus of neonatal rats. Conversely, inhibition of the PI3K/Akt pathway attenuated HO-1 expression and inhibited the neuroprotective effect of argon. The inhalative administration of argon also upregulated Nrf2 and NADPH dehydrogenase, quinone 1, and SOD. Blocking the PI3K/Akt pathway with wortmannin or the Erk1/2 pathway with the specific inhibitor U0126 also reduced the neuroprotective effect of argon [104,131]. Ulbrich et al. further demonstrated that Erk1/2 inhibition partially abolished Argon-induced reduction in caspase cleavage and HO-1 suppression (Figure 3) [102]. In a recent study, Scheid et al. demonstrated that preconditioning with 74 Vol% argon for 2 h protects human neuroblastoma cells from rotenone injury, suggesting that even prophylactic administration could be possible and opening up new fields of application [132].

Recently published studies provide evidence that argon has an impact on other systems as well. Kiss et al. showed improved cardiac function after cold ischemia and reperfusion [133]. Anesthetized Sprague Dawley rats were ventilated and received three intervals of argon (50 Vol%, O_2_ 21 Vol%, N_2_O 29 Vol%, 5 min) before the hearts were excised and evaluated in an erythrocyte-perfused isolated working heart system. After cold ischemia and reperfusion, the hearts of rats preconditioned with argon showed enhanced recovery of their cardiac output, stroke volume, and coronary blood flow compared with the control group. Additionally, the activation of JNK and expression of HMGB-1 were markedly reduced in left ventricle samples.

Several questions remain regarding the clinical use of argon. How are neuroscientists going to integrate argon in future clinical settings? Its protective effects after ischemia–reperfusion injury have been investigated in different in vivo models for different organ systems, and multiple effects have been determined on apoptotic and inflammatory pathways, including transcription factors, in the last decade (Table 1). Even surface molecules, such as receptors, seem to be part of the mechanism of action of argon. Although only partially suitable for clinical use in stroke, xenon has already been advanced to the next step and is now undergoing phase II clinical trials.

Another question is whether argon is ready for use in humans. Due to a lack of medical approval, argon remains far from routine clinical use, although it has been tested in large animal studies. Alderliesten et al. demonstrated hemodynamic safety in a porcine model, as argon at concentrations up to 80% did not change the heart rate, mean arterial blood pressure, regional cerebral saturation of oxygen, or electroencephalography traces [134]. By contrast, another porcine trial showed increases in systemic vascular resistance and a negative effect on cardiac output. Therefore, special care is advised in further test series, and cardiovascular monitoring is highly recommended [135,136]. Russian investigators have tested argon inhalation in healthy humans during physical exercise in a hypoxic environment and have reported increased oxygen consumption at argon concentrations up to 85%. However, no differences were detected in the peripheral capillary saturation measured using pulse oximetry [137]. Ristagno et al. are currently conducting the first clinical phase I trial investigating a potential beneficial effect of argon on neurological function in patients after sustained out-of-hospital cardiac arrest. The time has arrived to register argon as a medical gas, to allow its use in clinical studies and to implement argon in its first human trials [11]. The question of clinical feasibility when having to use higher noble gas concentrations in the air to achieve a neuroprotective effect, requiring a reduction in the inspired fraction of oxygen, has been discussed above for helium and xenon. Similar to xenon, there is some evidence suggesting a dose-dependent neuroprotective effect of argon, showing better results when the noble gas is applied in high inspiratory concentrations of up to 75%. However, an argon concentration of 50% has been described as effective in a number of studies [112,128]. Since respiratory insufficiency with the possible need for a higher fraction of inspired oxygen is common among stroke patients, especially those severely affected, the concentration of argon applied might be reduced while still achieving neuroprotection, albeit making it more difficult to tap its full neuroprotective potential. On the other hand, the necessity to reduce the fraction of inspired oxygen to accommodate for a sufficient noble gas percentage in the inhaled air might actually be a part of the neuroprotective effect of noble gases, since oxygen toxicity by overexposure may be prevented and oxidative stress by ROS may be reduced [105]. However, patients suffering from end-stage respiratory disease who constantly need oxygen support in very high concentration may not be eligible for argon treatment, since a therapeutic dosage cannot be safely applied in this patient cohort.

Further considerations regard the application of noble gases. Although all noble gases are technically compressible, during re-expansion a large amount of energy is released, making compressed noble gases potentially dangerous to use in a clinical context. To the best of the author’s knowledge, only the inhalative application of the pure substances is reasonable at this point. Its high production costs argue against the use of xenon. Argon, on the other hand, is readily available, non-combustible and cost-effective, making its routine clinical use more likely.

## 4. Conclusions

Cerebral injury secondary to stroke is a major cause of morbidity and mortality worldwide. Despite improved treatment options for revascularization, outcomes remain unfavorable, underlining the importance of adjuvant therapies. Several volatile agents have been shown to exert numerous neuroprotective effects in vitro and in various animal models of stroke. Volatile anesthetics are routinely used for anesthesia in humans; however, the available clinical data regarding their neuroprotective effect are inconclusive, indicating a need for further randomized controlled trials. Furthermore, the anesthetic properties limit their usability in the context of stroke treatment, as the ability to assess neurological function is paramount when navigating care for stroke patients. Despite promising experimental data, the neuroprotective potential of the gaseous molecules H_2_S and CO have not yet been investigated in clinical studies. Due to concerns of handler toxicity and a comparatively small therapeutic window, a future routine clinical use of either one appears to be unlikely at this point. Regarding noble gases, the neuroprotective potential of xenon is currently being evaluated in clinical trials, despite the possible disadvantages of this substance, including its anesthetic effects and high production costs. Conversely, both helium and argon do not act as anesthetic substances under normobaric conditions. Of the two, argon, due to its reproducible neuroprotective effect in various animal models, cost effectiveness, and safe handling, seems to show particular promise as a potential adjunct in the prophylaxis and therapy of stroke, and should, therefore, be further investigated in clinical trials. Volatile anesthetics and noble gases may present the future in treating neuronal disorders like stroke.

## Figures and Tables

**Figure 1 cells-12-02480-f001:**
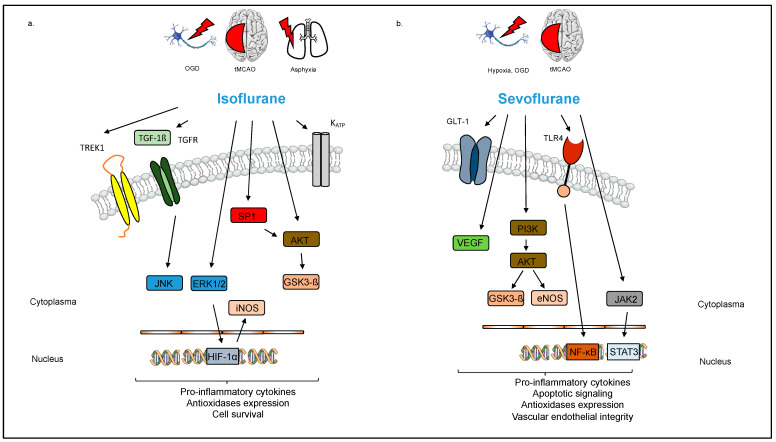
Schematic overview of neuroprotection by the volatile anesthetics sevoflurane and isoflurane. (**a**) Isoflurane promotes cell survival by activating the DNA binding complex HIF-1α, resulting in increased levels of Erk1/2 and iNOS. Isoflurane also attenuates apoptosis via activation of TREK1- and K_ATP-_channels. Activation of the TGF- β /JNK signaling pathway facilitates part of the neuroprotective effect of isoflurane. In addition, isoflurane leads to increased expression of SP1 and activation of the PI3K/Akt signaling pathway. Isoflurane, but not sevoflurane promotes part of its detrimental action with the p75-NTR receptor, located solely in neurons. (**b**) Sevoflurane reduces the expression of pro-inflammatory cytokines by regulating the TLR-4/NF-κB signaling pathway. Furthermore, sevoflurane improves neuronal survival and decreases apoptosis and cellular atrophy via the PI3K/Akt and JAK/STAT pathways. Sevoflurane also acts as an anti-oxidant and helps maintain vascular endothelial integrity.

**Figure 2 cells-12-02480-f002:**
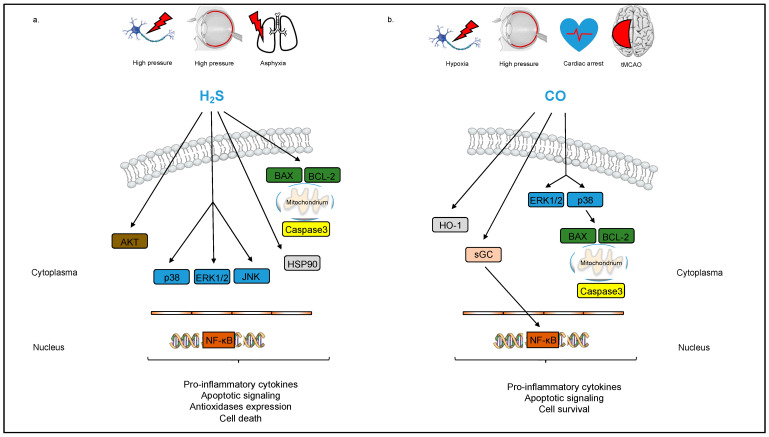
Schematic overview of neuroprotection by the gaseous molecules H_2_S and CO. (**a**) H_2_S suppresses apoptosis of neuronal cells by reducing the expression of the pro-apoptotic Bax protein and increasing the expression of the anti-apoptotic protein BCL-2. Regulation of the p38/Erk1/2 signaling pathway and the heat-shock response are key elements in facilitating the neuroprotective effect of H_2_S. Furthermore, H_2_S decreases the expression of apoptotic and pro-inflammatory markers, such as caspase-3, ICAM-1, VEGF, and iNOS. (**b**) CO reduces neuroinflammation by downregulation of the expression of pro-inflammatory cytokines. CO also differentially regulates the activation of MAP kinases ERK1/2 and p38, resulting in anti-apoptotic signaling. Moreover, CO promotes neuronal cell survival and modulates the heat-shock response.

**Figure 3 cells-12-02480-f003:**
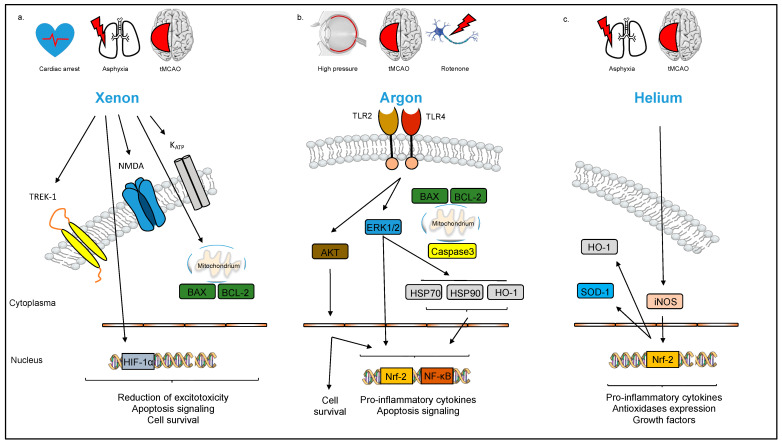
Schematic overview of noble gas-induced neuroprotection. (**a**) Xenon’s neuroprotective effect is mediated by the inhibition of NMDA receptors. In addition, Xenon can activate potassium channels (TREK-1) and functions as a potassium channel opener leading to increased K_ATP_ currents. Targeting Bax, Bcl-2 and HIF-1α, Xenon also provides anti-apoptotic and anti-inflammatory effects. (**b**) TLR2 and 4 are responsible for argon-mediated neuroprotective effects. The downstream signaling pathways PI3K/Akt and ERK1/2 impact transcription factors promoting cell survival and reducing pro-inflammatory cytokines and apoptosis signaling. (**c**): Helium promotes neuroprotection mainly via nitric oxide (NO) and increases the expression of antioxidases such as SOD-1 and HO-1.

## Data Availability

Not applicable.

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
