# Peer review of "Neuroprotection Is in the Air—Inhaled Gases on Their Way to the Neurons"

_cells, 2023, doi:10.3390/cells12202480_

Round 1

Reviewer 1 Report

The review entitled “Neuroprotection is in the air- inhaled gases on their way to the neurons” is a well-designed work that uses Standard English and is written in an interesting and appropriate order of issues. The authors made an excellent description of the intervention of different gases (volatile anesthetics, carbon monoxide, and noble gases) as neuroprotectors in cerebral injury.

My comment concerns the missing detailed data. For example,

Line 218  “...greater exposure to inhalation anesthetics…” There is no definition of greater exposure.

Line 430 “…Late pre-conditioning with 70% helium significantly decreased the infarct ratio...”

The values are needed in all of The Review. It can be added to a Table that includes dosage, timing of the administration, experimental model used, etc.

Other important information that is missing is the percentage of protection at least for infarction (% reduction).

Line 468 “The effect of xenon is well studied and is mediated by the inhibition of NMDA receptors…”. There is information that indicates “that 50% atm xenon a model of hypoxia-ischemia when applied immediately after injury or after a delay of 3 h injury has a neuroprotective effect (Banks et al., 2010). I obtained this information from the abstract, therefore in the full paper important information can be found.

Banks P, Franks NP, Dickinson R. Competitive inhibition at the glycine site of the N-methyl-D-aspartate receptor mediates xenon neuroprotection against hypoxia-ischemia. Anesthesiology. 2010 Mar;112(3):614-22. doi: 10.1097/ALN.0b013e3181cea398. PMID: 20124979.

I suggest that the authors add this type of information in the text or in tables.

Reviewer 2 Report

1.      While the introduction states that inhaled gases seem promising, it only briefly mentions their potential efficacy based on animal studies. Some skepticism could be expressed here, given the jump from animal models to human treatment is significant.

2.      Good introductions often provide "signposts" that outline what the reader can expect from the remainder of the article. This introduction could benefit from a sentence or two summarizing the main points that will be covered.

3.      While the introduction covers a lot of ground, the flow could be improved for better readability and comprehension. The transition between discussing the limitations of current treatments to the promise of inhaled gases is somewhat abrupt.

4.      While the 2nd section covers various aspects of pathophysiology, it jumps from one mechanism to another quite quickly. A smoother flow or transitional sentences could help the reader better grasp how these mechanisms are interconnected.

5.      The section mentions that in later stages of cerebral ischemia, neuroinflammatory mechanisms can have protective properties. This is an interesting point but might confuse readers if not adequately explained or contextualized.

6.      The text mentions the neurotoxic effects of sevoflurane briefly but could benefit from a more extended discussion on the potential risks or downsides.

7.      While the text does mention some clinical outcomes, it's unclear how significant these are, or how they translate into clinical practice recommendations.

8.      Both Hydrogen Sulfide (H2S) and Carbon Monoxide (CO) have concentration-specific effects, making it difficult to generalize their protective roles. At higher concentrations, they can have toxic or lethal effects.

9.      The text discusses the effects of H2S and CO but doesn't compare them to existing treatments or interventions. It's unclear if these gaseous molecules offer advantages over current standard-of-care therapies.

10.  The use of poisonous gases for therapeutic applications would face significant regulatory scrutiny. This point is not adequately discussed in the text.

11.  Unlike other noble gases, neon has not been found to offer neuroprotection based on in vitro models. And Due to the lack of proven neuroprotective effects, the paper concludes that the potential of neon as a future adjuvant treatment for stroke is questionable.

12.  High doses of helium (70%) might be problematic for patients with respiratory issues or those needing high concentrations of oxygen.

13.  Xenon can antagonize 5-hydoxytryptamine type (5-HT)-3 receptors, potentially exacerbating symptoms like nausea and vomiting, a high concentrations of xenon (up to 75%) may be required for neuroprotection, raising concerns about oxygen levels for the patient.

14.  Expand a bit on the potential directions for future research and clinical trials involving these gaseous molecules. Are there specific patient populations or stages of stroke that might benefit most? Are there innovative strategies for administration that could be explored?

Reviewer 3 Report

The subject of your manuscript is interesting, but without new information.

1.     In the introduction you talked too much about the mechanism of neuronal cell death and too little about the gases used to prevent/alleviate it.

2.     You have not mentioned anything about the aim and objectives of this manuscript.

3.     I recommend making a table about the role of noble gases similar to the one made with anesthetics.

4.     The conclusion is too long. I recommend condensing the information into a single paragraph.

Moderate editing of English language required

Round 2

Reviewer 1 Report

The submission of the final version of the manuscript was changed according to my comments. My concerns were properly addressed, therefore, I consider that this manuscript should be published.

Reviewer 2 Report

Thank you for taking all comments into consideration, the manuscript is much better now.